# Medication-Related Osteonecrosis of the Jaws (MRONJ) in Children and Young Patients—A Systematic Review

**DOI:** 10.3390/jcm12041416

**Published:** 2023-02-10

**Authors:** Hemil Dario Rosales, Henry Garcia Guevara, Stefania Requejo, Maria Dianella Jensen, Julio Acero, Sergio Olate

**Affiliations:** 1Graduate Program in OMFS, Hospital General del Oeste, Caracas 1030, Venezuela; 2Department of Oral Surgery, Santa Maria University, Caracas 1073, Venezuela; 3Division of Oral and Maxillofacial Surgery, Hospital Ortopedico Infantil, Caracas 1050, Venezuela; 4Department of Oral and Maxillofacial Surgery, Hospital Ramon y Cajal, 28034 Madrid, Spain; 5Center of Excellence in Morphological and Surgical Studies (CEMyQ), Universidad de La Frontera, Temuco 4780000, Chile; 6Division of Oral, Facial and Maxillofacial Surgery, Universidad de La Frontera, Temuco 4780000, Chile

**Keywords:** osteonecrosis, antiresorptive, MRONJ

## Abstract

Medication-related osteonecrosis of the jaw (MRONJ) is defined by the American Association of Oral and Maxillofacial Surgeons (AAOMS) as the presence of an exposed bone area in the maxillofacial region, present for more than eight weeks in patients treated with the use of antiresorptive or antiangiogenic agents, with no history of radiation or metastatic disease. Bisphosphonates (BF) and denosumab (DS) are widely used in adults for the management of patients with cancer and osteoporosis, and recently there has been an increase in their use in child and young patients for the management of disorders such as osteogenesis imperfecta (OI), glucocorticoid-induced osteoporosis, McCune-Albright syndrome (MAS), malignant hypercalcemia, and others. There are differences between case reports in adults compared to child and young patients related to the use of antiresorptive/antiangiogenic drugs and the development of MRONJ. The aim was to analyze the presence of MRONJ in children and young patients, and the relation with oral surgery. A systematic review, following the PRISMA search matrix based on the PICO question, was conducted in PubMed, Embase, ScienceDirect, Cochrane, Google Scholar, and manual search in high-impact journals between 1960 and 2022, publications in English or Spanish, including randomized and non-randomized clinical trials, prospective and retrospective cohort studies, cases and controls studies, and series and case reports. A total of 2792 articles were identified and 29 were included; all of them published between 2007 and 2022, identifying 1192 patients, 39.68% male and 36.24% female, aged 11.56 years old on average, using these drugs mainly for OI (60.15%); 4.21 years on average was the therapy time and 10.18 drug doses administered on average; oral surgery was observed in 216 subjects, reporting 14 cases of MRONJ. We concluded that there is a low presence of MRONJ in the child and youth population treated with antiresorptive drugs. Data collection is weak, and details of therapy are not clear in some cases. Deficiencies in protocols and pharmacological characterization were observed in most of the included articles.

## 1. Introduction

Bisphosphonates (BFs) are a group of antiresorptive drugs widely used in adults with pathological conditions, such as Paget’s disease, multiple myeloma, and conditions associated with cancer such as bone metastasis and hypercalcemia of malignancy [1,2,3,4,5,6,7,8,9,10,11,12,13].

The potential of BF therapy to improve survival rates remains controversial; however, its positive effect on the quality of life of patients with advanced bone cancer has been demonstrated [14]. Oral medications are also used for the management of osteoporosis and osteopenia [15,16].

BFs, especially those belonging to the nitrogen-containing subset, decrease bone resorption through inhibition of the enzyme farnesyl diphosphate synthase in the mevalonate pathway [17]. The mechanism is thought to involve interruption of the osteoclast cytoskeleton, impaired intracellular vesicular function, increased apoptosis, and decreased osteoclastic function [18,19].

In recent years, BFs have been reported to be associated with an increased risk of developing osteonecrosis of the jaws. The first description was reported in 2003 [20]; the most relevant report was presented by Marx [21]. Despite the fact that the first cases of osteonecrosis of the jaws were reported more than 15 years ago [21], the pathophysiology and mechanism is not completely clarified [22,23,24,25]. Hypotheses have been proposed that attempt to explain the unique location of this entity, exclusive to the jaws, including change in bone remodeling, over-suppression of bone resorption, inhibition of angiogenesis, frequent micro-trauma, suppression of innate or acquired immunity, vitamin D deficiency, soft tissue toxicity, inflammation, or infection [14].

On the other hand, in addition to BFs, denosumab (DS), which is a highly specific human monoclonal antibody (IgG2) against the human receptor activator of nuclear factor kappa-Β ligand (RANK Ligand), is also associated with the appearance of osteonecrosis of the jaws in a similar way to bisphosphonates [26,27,28].

In addition to antiresorptive drugs (BF and DS), there are agents from the group of antiangiogenic drugs related to osteonecrosis of the jaws, such as bevacizumab, sirolimus, sunitib, and others [29].

MRONJ is defined by the AAOMS as the presence of an area of exposed bone or the possibility of probing bone tissue through an intraoral or extraoral fistula in the maxillofacial region, present for more than eight weeks, in a patient with current or earlier treatment with antiresorptive or antiangiogenic agents, with no history of radiation therapy or metastatic disease in the jaws [14].

BF and DS are widely used in adults, and recently there has been an increase in their use in children and adolescents for the management of OI, glucocorticoid-induced osteoporosis, MAS, malignant hypercalcemia, or others [30,31].

There is an important difference between reports of MRONJ in adults compared to pediatric and young adolescent patients, even though these drugs are widely used in this population.

Currently, there are few reports in children and adolescents showing the relationship between OI, malignant pathology, osteoporosis, among others, and treatment with antiresorptive/antiangiogenic agents. The risk of developing MRONJ in such patients is possible; additionally, the influence of dental surgical treatments must be estimated.

The aim of this systematic review is to analyze the incidence of MRONJ in children and adolescents, and the relations with a history of invasive dental treatment.

## 2. Materials and Methods

In this systematic review, the PICO [32] question was established: Does MRONJ represent a risk in pediatric and adolescent patients treated with antiresorptive/antiangiogenic drugs in various pathologies? The systematic review performed using the PRISMA Statement (Preferred Reports for Systematic Reviews and Meta-analyses) as a guide [33,34].

The following inclusion criteria were considered: articles in English and Spanish, with child–youth population, undergoing treatment with antiresorptive and/or antiangiogenic drugs using intravenous route (IV) or oral route (po), with reports of cases of MRONJ. Case reports and case series, prospective and retrospective associated studies, case-control studies, randomized and non-randomized clinical trials were included. The exclusion criteria were literature review, treatment in subjects aged more than 18 years old, and the inability to access the full text.

In the first round of the search, abstracts were reviewed and all articles containing keywords were retained. Full versions were obtained for all the articles that met the inclusion criteria. In the second round of the search, a manual analysis of the references in each article was realized. A search for unpublished literature or data was not performed. Case reports and case series were also evaluated, in order to identify cases. The electronic search was complemented by a manual search as previously mentioned. In the third search round, each article was critically reviewed for validity and bias assessment and the following data were extracted: title, authors, journal, year of publication, type of study, total of pediatric and adolescent patients, total of pediatric and adolescent patients under antiresorptive/antiangiogenic treatment, sex, average age in years (range), indication for antiresorptive/antiangiogenic therapy, medication used, average time of use of the drug in years (range), number of average doses (range), data on dosage, route of administration, cumulative dose, adjunct medication therapy, MRONJ reports, comorbidities, MRONJ detection method, use of antimicrobial therapy.

The literature search strategy was carried out independently by two established evaluators (R.H and G.H) following the same search pattern. In case of disagreement during any stage of the review process, a third evaluator was included [32,33].

The following databases were incorporated into the systematic search: PubMed, Embase, ScienceDirect, Cochrane, Google Scholar, considering the literature from 1990 to April 2022. Manual search was included in this protocol, including articles from the area published in journals of high impact in selected languages (Journal of Oral and Maxillofacial Surgery, British Journal of Oral and Maxillofacial Surgery, Journal Of Neurosurgery, Asian Journal of Oral and Maxillofacial Surgery, Revista Española de Cirugía Oral y Maxilofacial, Revista Medicina Oral Patología Oral Cirugía Bucal). The following search terms were used: “Osteonecrosis jaw” or “ONJ” or “BRONJ” or “MRONJ” + “children” or “young” or “pediatric” or “paediatric” + “bisphosphonates” or “denosumab” or “antiangiogenics” + “osteogenesis imperfecta” or “Paget disease” or “fibrous dysplasia” or “cancer” in English, and “Osteonecrosis maxilar” o “ONM” o “OMAB” o “OMAM” o “ONMAM” + “niños” o “jóvenes” o “pediátrico” + “bifosfonatos” o “denosumab” o “antiangiogénicos” + “osteogenesis imperfecta” o “enfermedad de Paget” o “displasia fibrosa” o “cáncer” in Spanish.

## 3. Results

### 3.1. Selection

A total of 2792 articles were obtained in the five databases. After analysis of duplicate articles, 2417 were excluded. Another 309 articles were excluded in the title evaluation, and an additional 17 in the abstract review, making a total of 48 articles eligible for full-text evaluation; 1 article was added from the manual search. After the full-text review, 29 articles were included in the study that met the objectives of the systematic review [35,36,37,38,39,40,41,42,43,44,45,46,47,48,49,50,51,52,53,54,55,56,57,58,59,60,61,62,63] (Figure 1) (Table 1 and Table 2).

### 3.2. Years of Publication and Types of Study

Articles published between 2007 and 2022 were included, these being 16 cohort studies (55.17%), 9 case series (31.03%), 2 case reports (6.89%), 1 randomized clinical trial (3.44%), and 1 non-randomized clinical trial (3.44%).

### 3.3. Patients by Sex

The total number of children and adolescents among the 29 articles was 1595; of these, 1192 received antiresorptive/antiangiogenic medication and were evaluated in search of MRONJ, of which 473 (39.68%) were male, 432 (36.24%) were female, and 287 (24.07%) gave no data about sex.

### 3.4. Patients by Age

In total, 89.65% (*n* = 26) of the articles presented data to calculate the average age, resulting in 11.56 years for 1031 patients (range 0.01 to 32.00 years). It is important to show that in 31.03% of the articles there are patient older than 20 years. In all, 10.34% (*n* = 3) of the articles did not provide data for the analysis of age [43,55,56].

### 3.5. Antiresorptive/Antiangiogenic Medication Used

Reported was the use of pamidronate (PD) (*n* = 451; 37.83%), zoledronate (ZD) (*n* = 257; 21.56%), neridronate (ND) (*n* = 177; 14.84%), unspecified between PD and ZD (*n* = 156; 13.08%), PD plus ZD (*n* = 37; 3.10%), ibandronate (IB) (*n* = 20; 1.67%), unspecified between PD, alendronate (AL), and ZD (*n* = 18; 1.51%), AL (*n* = 12; 1.00%), DS (*n* = 8; 0.67%). In 56 patients (4.69%), the involved drug was not specified [51,55,58]. Antiangiogenic drugs were not used in any patient.

### 3.6. Conditions for Antiresorptive Therapy

Osteogenesis imperfecta (OI) (*n* = 717; 60.15%) was the main reason for the use of these drugs. Other conditions were related to: osteoporosis/low bone mineral density (*n* = 203; 17.03%), malignant pathology (*n* = 153; 12.83%), fibrous dysplasias/bone dysplasias/MAS (*n* = 46; 3.85%), avascular femoral necrosis (*n* = 37; 3.10%), major prostatic thalassemia (*n* = 12; 1.00%), neuromuscular disorders (*n* = 11; 0.92%), giant-cell bone tumor (*n* = 8; 0.67%), rheumatic disorders (*n* = 3; 0.25%), Crohn’s disease (*n* = 1; 0.08%), and transverse myelitis (*n* = 1; 0.08%).

### 3.7. Drug Usage Time

A total of 27.58% (*n* = 8) of the articles did not show data to calculate the time involved in the use of antiresorptive therapy [35,36,43,50,51,53,54,55]. On the other hand, 68.96% (*n* = 20) of the articles showed data for the time analysis, resulting in 4.21 years on average for 948 patients [37,38,39,40,41,42,44,45,46,47,48,49,52,56,58,59,60,61,62,63]. The lowest average of treatment time was 0.64 years [41] and the longest was 7.60 years [47].

### 3.8. Dosage and Administration

A total of 55.17% (*n* = 16) of the articles did not present any data to calculate the drug dosage or administration [38,39,40,43,47,48,49,51,52,54,55,57,58,59,60,62]. On other hand, 44.82% (*n* = 13) of the articles showed data, resulting in 10.18 drug dosage on average used in 476 patients [35,36,37,41,42,44,45,46,50,53,56,61,63]. The lowest average dosage was 4.16 [63] and the highest was 32.00 [61].

In total, 92.11%. of the patients (*n* = 1098) received the drug through intravenous route (IV), 1.51% (*n* = 18) using multiple routes of administration, 1.00% (*n* = 12) using the oral route (po), and 0.67% (*n* = 8) through subcutaneous route (SC). Three articles did not include data about the route of administration (*n* = 56; 4.69%) [51,55,58].

### 3.9. Posology

In total, 41.37% of the articles (*n* = 12) used PD. In five of these articles (17.24%), doses of 1 mg/kg were used, with different strategies, as follows: 1 mg/kg/dose every 2 months approximately [37], 1 mg/kg every month [38], 1 mg/kg/dose every 3 months [54], 1 mg/kg daily for 3 consecutive days repeating at between 4 and 6 months [60], and 1 mg/kg on the first day (maximum 30 mg), according to serum calcium on the second day: >2.2 mmol/L: 1 mg/kg (maximum 60 mg); between 2 and 2.2 mmol/L: 0.5 mg/kg (maximum 30 mg); <2 mmol/L: no more infusions, repeating every 3 months [63].

In all, 6.89% of the articles (*n* = 2) established schemes based on body surface area and quarterly dose increases as follows: monthly infusions of 10 mg/m^2^ the first 3 months, 20 mg/m^2^ the second 3 months to continue 30 mg/m^2^, adjusting the dose after 2 years in relation to response (assessed by bone densitometry, blood and urine bone turnover markers, and regression of vertebral fractures by compression) [47], and 10mg/m^2^ the first 3 months, 20 mg/m^2^ the second 3 months to continue 30–40 mg/m^2^ in monthly infusions [49]. In all, 3.44% of the articles (*n* = 1) established schemes by age groups: 0.5 mg/kg daily for 3 consecutive days every 2 months in children under 2 years old, 0.75 mg/kg daily for 3 consecutive days every 3 months in patients aged 2 to 3 years, and 1 mg/kg daily for 3 consecutive days every 4 months in patients older than 3 years with a maximum dose of 60 mg per day, maximum infusion concentration rate of 0.1 mg/mL, and duration of infusion administration of 3–4 h [39]. On the other hand, in 3.44% of the articles (*n* = 1), the following scheme was established according to the pathology: 9 mg/kg/year every 2–4 months for patients with primary osteoporosis and 4 mg/kg/year every 3 months for patients with secondary and glucocorticoid-induced osteoporosis [53]. In 6.89% of the articles (*n* = 2), the data are limited, only exposing that the last dose before dental surgery was 60 mg [50] and a single dose [57]. In 3.44% (*n* = 1) of the articles, data on dosage or administration schedules of PD were not provided [40].

A total of 10.34% of the articles (*n* = 3) used ND with the following schemes: 2 mg/kg (maximum 100 mg) every 3 months for 3 years, [42] one dose every 3 months without other data [43], and 1 mg/kg/dose day for 2 consecutive days every 3–4 months in patients younger than 1 year, and 2 mg/kg/dose in a single session every 3–6 months in the rest of the patients [48].

In total, 6.89% of the articles (*n* = 2) used AL. In one of them, the following schedule was applied: 35 mg once a week [57]. The other article lacks data about dosage and administration schedules [38]. In all, 3.44% of the articles (*n* = 1) used IB, at a dose of 2 mg every 3 months [46].

In total, 37.93% of the articles (*n* = 11) used ZD. In all, 6.89% (*n* = 2) did not provide data about drug dosage [38,40]. On the other hand, 6.89% (*n* = 2) of the articles [44,59] used doses of 0.0125 mg/kg (first dose) and 0.025 mg/kg (second dose), administered 6 weeks after the first dose, with a third dose of 0.025 mg/kg being added 12 weeks after the first dose in one of the articles [44]. The schedules for both articles included next doses of 0.025 mg/kg with a 12-week interval. Additionally, 10.34% (*n* = 3) of the articles applied schemes according to age. The schemes included the administration of 4 mg approximately every 28 days for those over 10 years of age and 0.08 to 0.16 mg/kg approximately every 28 days for those under 10 years of age, [35] 2 mg for those under 1 year of age and 4 mg for those over 1 year of age every 3 months [45], and 10 monthly doses of 4 mg for those over 25 years of age; 10 monthly doses of 0.05 mg/kg the first two cycles and 4 mg the remaining eight cycles for patients between 18 and 25 years old; 10 monthly doses of 0.05 mg/kg in all cycles without exceeding 4 mg for patients under 18 years [56]. The rest of the articles where ZD was used applied the following schemes: 0.05 mg/kg every 6 months (maximum 4 mg per dose), [62] 0.04–0.05 mg/kg/dose every 4 months approximately, [37] scheme of induction doses between weeks 1 and 12 of treatment by cohorts: 1.2 mg/m^2^ (maximum 2 mg); 2.3 mg/m^2^ (maximum 4 mg); 3.5 mg/m^2^ (maximum 6 mg); according to tolerance, dose levels were scaled, adding a fourth cohort at a dose of 2.3 mg/m^2^ (maximum 4 mg) after determining the maximum tolerated dose [41] and 0.1 mg/kg/year every 6 months [53].

In total, 10.34% of the articles (*n* = 3) used DS. The doses used in the three articles were 120 mg via subcutaneous with loading doses at 8 and 15 days, except for one article [61] that reports loading doses at 28 days, to then continue with doses every 4 weeks [36,61] or every 28 days [52]. In one article, a patient was treated with a dose of 70–100 mg (initial dose) and loads at 8, 15, 21, and 28 days, and then continued once a month [36]. Conversely, 10.34% (*n* = 3) of the articles did not present data on dosage and posology [51,55,58].

### 3.10. Cumulative Dose

In total, 24.13% (*n* = 7) of the articles presented data about cumulative doses, in the following way: 19.8 mg/kg of PD corresponding to an equivalent dose in adults of 1190 mg and 0.49 mg/kg of ZD corresponding to adult equivalent dose of 29.4 mg [37], cumulative average dose of 40 mg/kg of PD (range 2.5 to 81 mg/kg) in patients who underwent tooth extraction, [39] 48 mg/kg of ZD on average (range 16 to 80 mg/kg) [45], 1679 mg average cumulative dose of ND (range 144 to 5307 mg); 50 mg/kg of ND on average (range from 10 to 100 mg/kg), [48] 1623 mg/m^2^ of PD on average (range of 140 to 4020 mg/m^2^), [49] 0.1 mg/kg of ZD per year on average [59], and 5520 and 2160 mg of DS for both patients study subjects [61].

Conversely, 75.86% of the articles (*n* = 22) did not present data on cumulative doses [35,36,38,40,41,42,43,44,46,47,50,51,52,53,54,55,56,57,58,60,62,63].

### 3.11. Concomitant Medication

In total, 51.72% of the articles (*n* = 15) reported the administration of vitamin D and calcium supplements [36,38,41,42,44,46,47,50,52,54,56,57,59,62,63]. In all, 13.79% (*n* = 4) of the articles reported the use of corticosteroids [36,37,38,57], 13.79% (*n* = 4) of the articles reported the use of antineoplastic agents [35,36,41,56], 6.89% (*n* = 2) of the articles reported the use of hormonal therapy [36,38], 6.89% (*n* = 2) of the articles reported the use of non-steroidal analgesics [40,63], 3.44% (*n* = 1) of the articles reported the use of antidepressants [40], 3.44% (*n* = 1) of the articles reported the use of proton pump inhibitors [40], and 3.44% (*n* = 1) reported the use of opiates (codeine and morphine) [63]. On the other hand, 6.89% of the articles (*n* = 2) express the non-use of cytostatic and steroids [49] and corticosteroids, chemotherapy, or bisphosphonates [61]. In all, 31.03% of the articles (*n* = 9) did not report data on other medications used [39,43,45,48,51,53,55,58,60].

### 3.12. Osteonecrosis of the Jaws

In total, 89.65% of the articles (*n* = 26) did not report cases of MRONJ or healing impairment in studies where dental procedures were included [35,37,38,39,40,41,42,43,44,45,46,47,48,49,50,52,53,54,55,56,57,58,59,60,62,63]. In 10.34% (*n* = 3) of the articles was reported: 1 case of impaired healing after exploratory maxillary surgery in a patient diagnosed with central giant-cell granuloma, which the authors correlated with stage 2 of MRONJ [36], 12 cases of MRONJ including thickening of the lamina dura in seven patients, full-thickness sclerosis in six patients, sclerotic changes in the mandibular canal in three patients, poorly healed or non-healed post-extraction socket and periapical radiolucency in five patients, widening of the periodontal ligament and osteolysis in four patients, bone sequestration in three patients, oroantral fistulas in two patients, widening of soft tissue and mild periosteal reaction in one patient. The mandible was involved in nine patients and the maxilla in three patients [51]; one case, after 44 doses of DS, showed a fracture of a lower molar by a sports accident, requiring tooth extraction. The risk of developing MRONJ was discussed. The risks of relapses of giant-cell bone tumor vs MRONJ were evaluated, and it was decided to allow the continuity of DS treatment after complete healing of the mucosa without bone exposure. The patient returned 2 months later, with acute pain in the post-extraction tooth socket and the presence of exposed bone tissue (stage 2 MRONJ according to the American Association of Oral and Maxillofacial Surgeons classification). DS treatment was stopped after a total of 46 doses (cumulative dose 98 mg/kg). Cultures of the area showed *Streptococcus milleri* and alpha-hemolytic *Streptococcus.* Non-surgical treatment of MRONJ was performed at the beginning (amoxicillin, metronidazole, and mouth washes) without success. Debridement and sequestrectomy were subsequently performed showing a moderate amount of necrotic bone tissue around the post-extraction socket with complete recovery after the surgery [61].

### 3.13. MRONJ Detection Method

In total, 55.17% of the articles (*n* = 16) do not show data on the MRONJ detection method [35,36,38,40,41,42,44,45,46,47,50,55,56,57,62,63]. In 37.93% of the studies (*n* = 11) were reported the use of clinical evaluation for the confirmation of MRONJ [37,39,43,48,49,50,51,54,58,59,60,61], and additionally, imaging exploration was used in 17.24% of articles (*n* = 5) [37,43,49,58,60]. In one article (3.44%), the search in notes of follow-up by the physician in the medical records was used [39].

### 3.14. Invasive Procedures

Invasive dental procedures were carried out in 37.93% of the articles (*n* = 11) [36,37,39,43,48,49,50,54,55,58,61] including 216 patients (18.12% of all patients) with 13.17 years old in average [36,37,39,49,50,54,61]; 30.09% (*n* = 65) were male, 21.75% (*n* = 47) were female, and 48.14% (*n* = 104) gave no information about sex. PD was used in 63.42% of patients (*n* = 137), ND in 10.18% (*n* = 22), unspecified between PD and ZD in 5.09% (*n* = 11), DS in 0.92% (*n* = 2). The drug is not specified in 20.37% of patients (*n* = 44) (Table 3).

A total of 545 invasive procedures were performed: primary tooth extractions (*n* = 371; 68.07%), permanent tooth surgical extractions (*n* = 123; 22.56%), periodontal treatment (*n* = 35; 6.42%), frenectomy (*n* = 8; 1.46%), endodontic treatments (*n* = 3; 0.55%), orthognathic surgery (*n* = 1; 0.18%), odontoma excision (*n* = 1; 0.18%), implant surgery (*n* = 1; 0.18%), exploratory maxillary surgery (*n* = 1; 0,18%), and surgical exposure of canine included (*n* = 1; 0.18%). In total, 79.08% (*n* = 431) of the procedures were performed during antiresorptive therapy, 19.08% (*n* = 104) were performed after the end of therapy, and in 1.83% (*n* = 10), the time for dental treatment is not clear.

In total, 36.36% of the articles (*n* = 4) showed data on the time using the antiresorptive treatments prior to invasive procedures: 0.66 years on average [36], 3.50 years on average for patients under treatment with PD, and 1.20 years on average for patients under treatment with ZD [37], 4.60 years on average [39], and 3.60 years on average [49]. In all, 27.27% of the articles (*n* = 3) presented data on cumulative doses, these being the following: 1250 mg on average for patients under treatment with PD and 11.10 mg on average for patients under treatment with ZD [37], 40 mg/kg on average [39] and 98 mg/kg [61].

In total, 45.45% of the articles (*n* = 5) presented data about use of antibiotic therapy, with two articles (18.18%) showing a treatment related to: amoxicillin 50 mg/kg administered 30–60 min before surgery, clindamycin 20 mg/kg 30–60 min before surgery in allergic patients, used in two patients [39] and without the use of antibiotic therapy [50].

In total, 36.36% of the articles [39,50,54] presented data on the postoperative follow-up time, these being 1.58 years of average postoperative follow-up. In one article (11.11%) with 2.48 years follow-up, the postoperative control was performed calling all the subjects or parents [39].

In 81.81% (*n* = 9) of the articles, no cases of MRONJ were reported [37,39,43,48,49,50,54,55,58]. In 18.18% of the articles, two cases of MRONJ were reported [36,61].

### 3.15. Risk of Bias Assessment

Due to the presence of different types of study, it was necessary to apply different checklists [64,65,66,67], in order to check the quality of the articles (Table 4).

## 4. Discussion

The time involved in the antiresorptive therapy is a risk factor for development of MRONJ [14]. Among patients with malignant pathology exposed to denosumab and zoledronate, the incidence of MRONJ increases directly proportional to the years of treatment, reaching a plateau between 2 and 3 years of treatment. The incidence of MRONJ is established at 1.3% for denosumab and 1.88% for zoledronate after 3 years of therapy [68]. For orally administered medication, patients without MRONJ have an average duration of treatment of 3.5 years, while patients who developed MRONJ have an average of 4.4 years [69,70]. Within the results of this systematic review, 27.58% of the articles failed to show data to calculate the duration of treatment [35,36,43,50,51,53,54,55]. In contrast, 68.96% of the articles present data that allow us to calculate the average duration of antiresorptive therapy as 4.21 years [37,38,39,40,41,42,44,45,46,47,48,49,52,56,58,59,60,61,62,63]. We can think that this group of patients show higher risk of developing MRONJ.

Another important factor in estimating the risk of MRONJ is related to the pathology under treatment. For example, the prevalence of MRONJ in patients with malignant disease exposed to zoledronate is 1% (100 cases per 1000 patients); in contrast, the same population exposed to placebo have a risk of developing MRONJ 50 to 100 times higher [71,72,73,74], this risk being comparable to patients with cancer exposed to denosumab [21,68,75]. In patients with osteoporosis, the prevalence is 0.1% with an increase to 0.21% when the use of the drug is for more than 4 years [69,70]. If compared with cancer patients, the risk is 100 times lower [14]. Even though, within the population of this systematic review, there are patients with malignant pathology (12.83%) and osteoporosis (17.03%), the main indication of antiresorptive therapy in the child–youth population was osteogenesis imperfecta (60.15%). The differences in indications for antiresorptive therapy between the pediatric/adolescence population and adults represent a limitation for the comparison of risks.

Some drugs, when administered simultaneously with antiresorptives, represent an increased risk of developing MRONJ, such as corticosteroids [76,77] and antiangiogenics [78,79]. Within the results of this study, 31.03% of the articles did not report data on the use of concomitant medication [39,43,45,48,51,53,55,58,60]. In all, 13.79% of the articles reported the use of corticosteroids [36,37,38,57]. Of the 68.96% (*n* = 20) of the articles that reported the use of concomitant medication, only two articles reported cases of MRONJ (two patients), and for that reason, the use of concomitant medication does not appear to be a risk factor in the development of MRONJ.

In total, 55.17% of the articles (*n* = 16) in this systematic review do not present data on the MRONJ detection method [35,36,38,40,41,42,44,45,46,47,50,55,56,57,62,63]. In 37.93% of the articles (*n* = 11), the use of clinical evaluation for the detection of MRONJ was reported [37,39,43,48,49,50,51,54,58,59,60,61], and, additionally, an imaging was described in 24% of the articles (*n* = 5) [37,43,49,58,60]. However, they do not report data about the oral health status of patients prior to initiation and during antiresorptive therapy, and few studies describe the criteria for the diagnosis of MRONJ.

As previously stated, the risks of developing MRONJ are related to the pathology and drug. In this systematic review, some articles presented dosing schedules that were applied to treat different pathologies in the same way [35,37,39], and other articles observed the use of different antiresorptive drugs to treat the same pathology [37,38,40]. The data provided by these articles make it impossible to analyze the risk of MRONJ in children and adolescents, especially because not all antiresorptive drugs have the same relative potency. In the case of bisphosphonates, which constitute the largest group of antiresorptive drugs, these differences appear; for example, zoledronate is at least ten times more potent than IB, which in turn is a thousand times more powerful than etidronate [80]. This increase in potency and, in turn, in toxicity is due to the presence of nitrogen within the molecular chain of bisphosphonate [81,82,83]. This fact makes it essential that studies using bisphosphonates include information about the drug and the cumulative doses, in order to contribute to establishing risk groups in these populations. In this systematic review, 3 studies lack data on the drugs used [51,55,58], 6 studies do not present specific data on dosage [40,43,50,51,55,58], and 22 studies do not present data on cumulative doses [35,36,38,40,41,42,43,44,46,47,50,51,52,53,54,55,56,57,58,60,62,63].

The prevalence for the subjects included in this review is close to 1.1%. If we consider the results of Moeini et al. [51], the prevalence decreases to 0.16%. The analysis of bias is presented in Table 4 [64,65,66,67] and because we included articles with different methodological designs, the use of different tools was necessary.

According to AAOMS [14], dentoalveolar surgery is considered a major risk factor for the development of MRONJ. Several studies reported that among patients with MRONJ, dental extraction is a very common variable (52% to 61% of patients) [79,84,85]. In this review, 216 patients received 545 invasive treatments performed mostly during antiresorptive therapy, including dental extraction in primary and permanent dentition, surgery for odontoma, dental implant surgery, and orthognathic surgery. Only two cases of MRONJ (0.92% of patients) were reported after exploratory maxillary surgery in a patient with central giant-cell granuloma [36] and tooth extraction from a fractured molar resulting from a sport accident [61]. These data allow us to clarify that invasive treatments in this population do not appear to be a risk factor for MRONJ.

The pediatric skeleton presents a thicker overlying periosteum, a greater osteogenic potential, and a greater remodeling potential than adult bone [86]. This creates evident physiological differences between the pediatric and adult skeleton, preventing their direct comparison [37]. Growing bone is more porous than adult bone because Haversian canals occupy much more space within bone mass [87]. Additionally, there is more vascularization [88], which may the most important protection against MRONJ. These physiological differences make the calculation of cumulative doses more complex in the child–youth population [37]. Is important to note that recent studies showed that chronic exposure to zoledronic acid induced significant reduction in osteogenic differentiation in in vitro models and this fact can be included in the conditions for the treatment [89]. In this line, exosomes can help to reduce the risk of side effects in the treatment under antiresorptive drugs [90].

Growth rates are higher in children and adolescence compared to adults, demonstrated by levels of biochemical markers of bone resorption and apposition such as serum alkaline phosphatase, serum osteocalcin, pyridoline and deoxypyridoline, NTX and CTX of mature collagen type I, serum calcium, among others [91]. Maybe, the high bone turnover may serve as a compensatory factor in the face of complications related to the effects of antiresorptive therapy, even long-standing ones, reducing the half-life of antiresorptive drugs.

It is important to note that in some stages of the bone growth process there is a delay in the apposition of minerals, as well as an increase in cortical porosity, showing an increased risk of fractures in early adolescence [92,93]. On the other hand, osteoclastic cells have not decreased in child and adolescent subjects, in contrast to adults or the elderly [94]. In the same line, the presence of vitamin D and calcium is important in subjects under antiresorptive drug therapy and in children this fact can be easier to control than in the adult population [95].

Anatomical factors could be another protective factor, since the primary teeth present shorter and narrower roots with some root resorption at the time of the extraction, and sockets are smaller with less requirements for bone resorption; in addition, the alveolar process being in growth with active bone apposition [54].

It was described that the development of MRONJ in patients with osteoporosis is related to suppuration, use of bisphosphonate, tooth extraction, and anemia [96]. Others showed that dental extractions, dental implants, and apical or periodontal surgery are the main factors involved in MRONJ [97]; additional risk factors such as dental prostheses with poor adaptation, excessive chewing force, poor oral hygiene, periodontal disease, and morphological bone irregularities have been related to MRONJ [98]. These conditions are absent in children and adolescents, and the masticatory force is lower in this group, showing another protective condition against MRONJ [99]. In the same line, *Actinomyces* spp. show a role in the pathogenesis of MRONJ [100,101] and this type of biofilm show differences between the child and adult populations.

As reported in this review and in others previously published [102], the main problem to compare results between studies was the lack in standardization, process and route in drugs, posology, timing, administration, and dosage, making it complex to compare results. The lack in data, low prevalence, and differences in methodology make it difficult to perform a metanalysis. The latest position paper of the AAOMS [103] states that there are very limited data describing the occurrence of MRONJ in the pediatric population for osteogenesis imperfecta and other conditions. No cases of MRONJ with sufficient evidence have been reported so far.

Additionally, the role of surgical management in improving MRONJ at early stages has also become a topic of discussion in the literature; it has even been included in the latest position paper of the American Association of Oral and Maxillofacial Surgery [103].

Surgical intervention should be explored and presented as a treatment option in an attempt to slow disease progression with the recognition that early surgical intervention may predict beneficial outcomes for patients [104].

Active clinical and radiographic surveillance is critical in the nonsurgical management of patients with Stage 1, 2, and 3 diseases to monitor signs of disease progression. In patients who demonstrate failure of nonsurgical therapy, early surgical intervention is recommended. In patients with a progressive clinical or radiographic picture in the disease or more advanced disease at presentation, the use of surgery is recommended [103,105]. MRONJ resection should be performed without first instituting prolonged nonoperative measures. MRONJ represents a complex wound for which surgical therapy can be performed in a timely manner [10]. Although controversy exists between operative and non-operative therapies, surgical treatment of patients has shown maintenance of mucosal coverage, improvement in quality of life, and timely resumption of antiresorptive therapy for all stages of MRONJ [106]. Nonetheless, the lack of evidence in pediatric and young patients makes it impossible at the moment to have any recommendations of early treatment in this populations treated with the drugs discussed in this research.

## 5. Conclusions

There is a low presence of MRONJ in the child and youth population. Data collection is weak, and details of therapy are not clear in some cases. Deficiencies in protocols and pharmacological characterization were observed in most of the included articles. The biological and physiological conditions involved in bone growth and development, as well as the proper dental and oral conditions, could be the most important protective factor against MRONJ. For future articles, it is recommended that a proper collection of the data is performed, where the drugs, time of treatment, possible trigger procedures of MRONJ, treatment, and follow-up must be recorded so further studies can be developed properly.

## Figures and Tables

**Figure 1 jcm-12-01416-f001:**
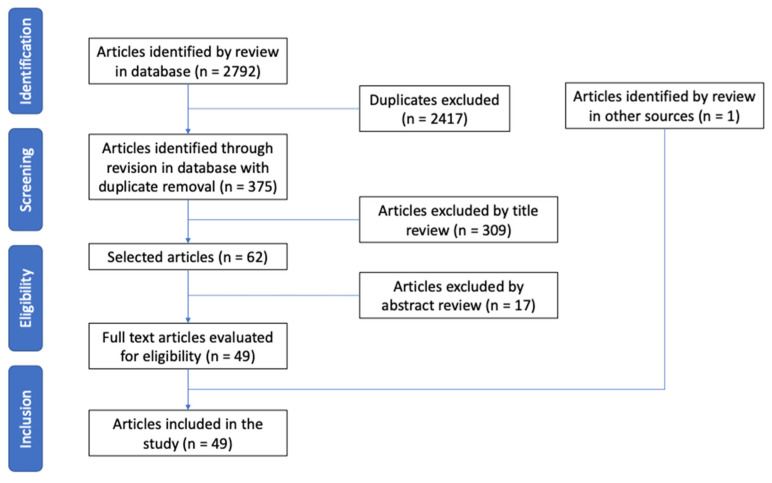
Flow chart for the selection of articles based on the PRISMA matrix.

**Table 1 jcm-12-01416-t001:** Distribution of variables of 29 articles related to MRONJ included in this systematic review.

Authors (Year)	Pts. under ART/AAT	Sex	Avg. Age in yr. (Range)	Indication for ART/AAT	Medication	Medication Use Time (Range)	Avg. Doses Number (Range)	Dosage Data	Cumulative Dose	Concomitant Medication
August et al. (2011) [35]	19	12 (M) 7 (F)	12.50 (1.10 to 23.10)	Malignant pathology	ZD	NDA	7.47 (1 to 21)	<10 yr.: 4 mg q28 days approx. >10 yr.: 0.08–0.16 mg/kg q28 days approx.	NDA	Vincristine, cyclophosphamide, doxorubicin, ifosfamide, etoposide, methotrexate, topotecan, temozolamide, gemcitabine, docetaxel, irinotecan, vinorelbine, busulfan, melfalan.
Bredell et al. (2017) [36]	5	2 (M) 3 (F)	18.00 (3.90 to 26.00)	CGCL	DS	NDA	14.20 (12 to 15)	Case 1: 70–100 mg SC once a wk. for the first mo. and then once a mo. Rest of cases: 120 mg SC initial dose, 120 mg on the 8th day and 15th day. Then 120 mg q4 wk.	NDA	Calcitonin, IFN alpha, vitamin D and calcium supplements, intralesional corticosteroids.
Brown et al. (2008) [37]	42	NDA	8.25	OI, MAS, osteoporosis, transverse myelitis	PD (*n* = 1) ZD (*n* = 4) PD + ZD (*n* = 37)	6.50	29.60	PD: 1 mg/kg/dose q2 mo. approx. ZD: 0.04–0.05 mg/kg/dose q4 mo. approx.	PD: 19.8 mg/kg corresponds to an equivalent dose in adults of 1190 mg. ZD: 0.49 mg/kg corresponds to an equivalent adult dose of 29.4 mg.	Corticosteroids in a single case.
Carpenter et al. (2007) [38]	18	11 (M) 7 (F)	11.30 (5.80 to 17.10)	Low BMD	PD. AL or ZD as alternatives	0.96	NDA	PD: 1 mg/kg q mo.	NDA	Growth hormone, sex hormones, prednisone, calcium and/or vitamin D supplements.
Chahine et al. (2008) [39]	278	136 (M) 142 (F)	14.70 (0.70 to 32.00)	OI, osteoporosis, fibrous/bone dysplasia, neuromuscular disorders, rheumatic disorders, Crohn’s disease	PD	4.60 (0.00 to 112)	NDA	<2 yr.: 0.5 mg/kg q day for 3 consecutive days q2 mo. 2–3 yr.: 0.75 mg/kg q day for 3 consecutive days q3 mo. >3 yr.: 1 mg/kg q day 3 consecutive days q4 mo. Maximum dose: 60 mg q day. Maximum infusion concentration: 0.1 mg/mL. Infusion administration duration: 3–4 h.	9 mg/kg (total annual dose). Avg. cumulative dose before tooth extraction: 40 mg/kg (2.5 to 81 mg/kg) in pts. who underwent tooth extraction and had records of PD Tx. (*n* = 45)	NDA
Feehan et al. (2018) [40]	33	18 (M) 15 (F)	9.00	OI	PD or ZD	7.00 (4 to 11.5)	NDA	NDA	NDA	Antidepressants (*n* = 6), proton pump inhibitor (*n* = 2), pain relievers (*n* = 1).
Goldsby et al. (2013) [41]	24	8 (M) 16 (F)	13.50 (7.00 to 22.00).	Malignant pathology	ZD	0.64	8.00	Induction dose (wk. 1 to 12): 1.2 mg/m^2^ (max. 2 mg) (*n* = 6); 2.3 mg/m^2^ (max. 4 mg) (*n* = 6); 3.5 mg/m^2^ (max. 6 mg) (*n* = 6). According to tolerance, the dose levels were scaled. A fourth group (*n* = 6) was added at a dose of 2.3 mg/m^2^ (max. 4 mg) after determining the maximum tolerated dose, in order to help assess the post-induction feasibility of ZD.	NDA	Cisplatin, adriamycin, methotrexate, ifosfamide, etoposide, calcium and vitamin D supplements.
Idolazzi et al. (2017) [42]	55	30 (M) 25 (F)	12.60 (5.00 to 19.00)	OI	ND	3.00	11.10 (3 to 13)	2 mg/kg (max. 100 mg) q3 mo. for 3 yr.	NDA	Calcium and vitamin D supplements.
Ierardo et al. (2017) [43]	20	12 (M) 8 (F)	NDA (8.00 to 14.00)	OI	ND	NDA	NDA	Dose q3 mo.	NDA	NDA
Johannesen et al. (2009) [44]	37	28 (M) 9 (F)	10.80 (6.01 a 1550)	AFN	ZD	1.18	6.70	1st dose: 0.0125 mg/kg. 2nd dose (at 6 wk.): 0.025 mg/kg. 3rd dose (12 wk. after the 1st dose): 0.025 mg/kg. Following dose: at 0.025 mg/kg 12 wk. apart.	NDA	Calcium and vitamin D supplements.
Kumar et al. (2016) [45]	26	NDA	7.00 (3.75 to 10.00)	OI	ZD	3.00 (0.91 to 5.08)	12,00 (3.66 to 20.3)	<1 yr.: 2 mg. >1 yr.: 4 mg q3 mo.	48 mg/kg on avg. (16 to 80 mg/kg).	NDA
Li et al. (2011) [46]	20	11 (M) 9 (F)	10.60	OI	IB	2.00	8.00	2 mg q3 mo.	NDA	Calcium and vitamin D supplements.
Lindahl et al. (2016) [47]	79	43 (M) 36 (F)	6.75 (0.10 to 17.10).	OI	PD	7.60	NDA	Monthly infusion of 10 mg/m^2^ (1st 3 mo.), 20 mg/m^2^ (2nd 3 mo.), then 30 mg/m^2^. After 2 yr., the dose was adjusted in relation to the response (evaluated by bone densitometry, bone turnover markers in blood and urine, and regression of vertebral compression fractures).	NDA	Calcium and vitamin D supplements.
Maines et al. (2012) [48]	102	47 (M) 55 (F)	12.26 (3.10 to 23.40)	OI	ND	6.81 (1.00 to 12.90)	NDA	Pts. who started treatment with less than 1 yr. (*n* = 15): Infusion 1 mg/kg/day for 2 consecutive days q3–4 mo. Rest of patients (*n* = 87): 2 mg/kg/dose in a single session q3–6 mo.	Cumulative avg. dose: 1679 mg (144 to 5307 mg). Cumulative avg. dose (per kg): 50 mg/kg (10 to 100 mg/kg).	NDA
Malmgren et al. (2008) [49]	64	NDA	8.10 (0.20 to 20.90).	OI	PD	4.50 (0.50 to 12.50)	NDA	1st 3 mo.: 10 mg/m^2^. 2nd 3 mo.: 20 mg/m^2^. Then: 30–40 mg/m^2^ in q mo. infusion.	1623 mg/m^2^ on avg. (140–4020 mg/m^2^).	Steroids or cytostatics were never administered.
Milano et al. (2011) [50]	1	1 (M)	4.66	OI	PD	NDA	8.00	Last dose before surgery: 60 mg.	NDA	Calcium and vitamin D supplements.
Moeini et al. (2013) [51]	12	3 (M) 9 (F)	13.00 (700 to 21.00)	Prostate thalassemia major	NDA	NDA	NDA	NDA	NDA	NDA
Naidu et al. (2014) [52]	1	1 (F)	9.00	CGCL	DS	1.50	NDA	120 mg q day for 28 days. Additionally, 2 loading doses were administered in 1mo. (8th and 15th day of Tx).	NDA	Calcium and vitamin D supplements.
Nasomyont et al. (2019) [53]	123	69 (M) 54 (F)	10.21 (0.01 to 20.60)	Osteoporosis	PD or ZD	NDA	5.27 (1 to 48)	Individualized doses for each case. On avg. PD was administered: 9 mg/kg/yr. q2-4 mo. (Primary osteoporosis group). 4 mg/kg/yr. q3 mo. (secondary and glucocorticoid-induced osteoporosis groups). ZD: 0.1 mg/kg/year every 6 months.	NDA	NDA
Ngan et al. (2013) [54]	1	1 (F)	12.00	OI	PD	NDA	NDA	1 mg/kg/dose q3 mo.	NDA	Calcium and vitamin D supplements.
Okawa et al. (2017) [55]	31	NDA	NDA	OI	NDA	NDA	NDA	NDA	NDA	NDA
Piperno-Neumann et al. (2018) [56]	110	NDA	NDA	Malignant pathology	ZD	0.83	10.00	>25 yr.: 10 monthly doses of 4 mg. 18–25 yr.: 10 monthly doses of 0.05 mg/kg for the first two cycles and 4mg for the remaining 8 cycles. Children <18 yr.: 10 monthly doses of 0.05 mg/kg in all cycles without exceeding 4 mg.	NDA	Calcium and vitamin D supplements. Chemotherapy regimens with: Methotrexate, etoposide, ifosfamide, cisplatin, doxorubicin.
Putman et al. (2018) [57]	14	NDA	14.70 (4.00 to 20.00)	Low BMD	AL (*n* = 12) y PD (*n* = 2)	1.90 (AL group)	NDA	AL: 35 mg po once a wk. (*n* = 12). PD: (single dose IV) (*n* = 2).	NDA	Vitamin D supplements, glucocorticoids.
Schwartz et al. (2008) [58]	13	9 (M) 4 (F)	9.10 (2 to 17.41)	OI	NDA	4.56	NDA	NDA	NDA	NDA
Simm et al. (2011) [59]	20	9 (M) 11 (F)	9.60 (3.30 to 16.50)	Osteoporosis	ZD	1.70 (0.50 to 2.00)	NDA	1st dose: 0.0125 mg/kg. 2nd dose: (at 12 wk.): 0.025 mg/kg. Following dose: (12 wk. after the previous dose): 0.025 mg/kg.	0.1 mg/kg per year on avg.	Calcium and vitamin D supplements.
Tessaris et al. (2016) [60]	13	6 (M) 7 (F)	20.30 (7.00 to 27.00)	Fibrous dysplasia, MAS.	PD	2.50	NDA	1 mg/kg/day for 3 consecutive days (1 daily infusion) repeating at intervals of 4–6 mo.	NDA	NDA
Uday et al. (2018) [61]	2	1 (M) 1 (F)	14.85 (14.00 to 15.70)	CGCL	DS	2.45	32.00 (18.00 to 46.00)	120 mg on days 1, 8, 15, and 28. Then q4 wk. (2.1 and 2.6 mg/kg/dose, respectively).	5520 and 2160 mg, respectively.	Corticosteroids, chemotherapy, and bisphosphonates were not administered to patients.
Vuorimies et al. (2011) [62]	17	8 (M) 9 (F)	10.10 (1.50 to 16.80)	OI	ZD	1.90 (1.00 to 3.20)	NDA	1 infusion q6 mo. at a dose of 0.05 mg/kg (maximum 4.0 mg daily).	NDA	Calcium and vitamin D supplements.
Wagner et al. (2011) [63]	12	9 (M) 3 (F)	10.84 (2.20 to 14.50)	Osteoporosis	PD	1.00	4.16 (1.00 to 21.00)	Protocol of 1 or 3 days depending on the healthcare center. 1st day: 1 mg/kg (max. 30mg). 2nd day: (according to serum calcium) >2.2 mmol/L: 1 mg/kg (maximum 60 mg). Between 2 and 2.2 mmol/L: 0.5 mg/kg (maximum 30 mg). <2 mmol/L: no more infusions. It is repeated q3 mo.	NDA	Calcium and Vitamin D supplements, acetaminophen, codeine and morphine.

AAT: antiangiogenic; AFN: avascular femoral necrosis; AL: alendronate; approx.: approximate; ART: antiresorptive; Avg.: average; BMD: bone mineral density; CGCL: central giant-cell lesion; DS: denosumab; F: female; IB: ibandronate; IV: intravenous; M: male; MAS: McCune-Albright syndrome; max.: maximum; mo.: month(s); ND: neridronate; NDA: No data available; OI: osteogenesis imperfecta; PD: pamidronate; po: per os (by mouth); pts.: patient(s); q: every; SC: subcutaneous; tx.: treatment; wk.: week(s); yr.: year(s); ZD: zoledronate.

**Table 2 jcm-12-01416-t002:** Presence of MRONJ and comorbidities in the articles included in this systematic review.

Authors (Year)	MRONJ Reported Cases	Comorbidities	MRONJ Detection Method	Use of Antimicrobial Therapy
August et al. (2011) [35]	No cases were reported.	NDA	Not specific.	N/A
Bredell et al. (2017) [36]	1 case (case 2). MRONJ stage 2 (poor healing after exploratory surgery).	NDA	Not specific.	N/A
Brown et al. (2008) [37]	No cases were reported.	NDA	Clinical and radiographic evaluation.	NDA
Carpenter et al. (2007) [38]	No cases were reported.	NDA	Not specific.	N/A
Chahine et al. (2008) [39]	No cases were reported.	NDA	Search in medical record and follow up control notes from dentist that treats patient.	Prophylactic antibiotic therapy was used in 12 patients.
Feehan et al. (2018) [40]	No cases were reported.	Psychiatric (*n* = 4), cardiovascular (*n* = 3), respiratory (*n* = 1), endocrinology (*n* = 1) ENT (*n* = 3), others (*n* = 4).	Not specific.	N/A
Goldsby et al. (2013) [41]	No cases were reported.	NDA	Not specific.	N/A
Idolazzi et al. (2017) [42]	No cases were reported.	NDA	Not specific.	N/A
Ierardo et al. (2017) [43]	No cases were reported.	NDA	Radiographic and clinical evaluation.	Amoxicillin 50 mg/kg 30–60 min before surgery. Clindamycin 20 mg/kg was used in allergy sufferers 30–60 min before surgery.
Johannesen et al. (2009) [44]	No cases were reported.	NDA	Not specific.	N/A
Kumar et al. (2016) [45]	No cases were reported.	NDA	Not specific.	N/A
Li et al. (2011) [46]	No cases were reported.	They do not present rickets, hyperparathyroidism, other hereditary or metabolic bone pathologies, history of treatment with bisphosphonates, abnormal renal function (exclusion criteria).	Not specific.	N/A
Lindahl et al. (2016) [47]	No cases were reported.	NDA	Not specific.	N/A
Maines et al. (2012) [48]	No cases were reported.	NDA	Clinical examination by dental surgeon.	NDA
Malmgren et al. (2008) [49]	No cases were reported.	Its absence is presumed due to the non-administration of steroids or cytostatics.	Clinical and radiographic reviews every 6 months by a dentist or doctor. Radiographic evaluation in two children younger than 3 years was omitted.	Penicillin for 7 days in one case to treat a postoperative infection, which healed without complications.
Milano et al. (2011) [50]	No cases were reported.	NDA	Clinical evaluation.	Ampicillin 50 mg/kg single dose prior to surgical procedure.
Moeini et al. (2013) [51]	12 cases.	NDA	Clinical evaluation.	N/A
Naidu et al. (2014) [52]	No cases were reported.	NDA	Not specific.	N/A
Nasomyont et al. (2019) [53]	No cases were reported.	NDA	Not specific.	N/A
Ngan et al. (2013) [54]	No cases were reported.	NDA	Clinical evaluation 1st month and 3rd month. There were no signs of infection, exposed bone, or osteonecrosis.	Amoxicillin for 5 days.
Okawa et al. (2017) [55]	No cases were reported.	NDA	Not specific.	NDA
Piperno-Neumann et al. (2018) [56]	No cases were reported.	NDA	Not specific.	N/A
Putman et al. (2018) [57]	No cases were reported.	NDA	Not specific.	N/A
Schwartz et al. (2008) [58]	No cases were reported.	NDA	Clinical evaluation in some cases.	13–15 patients (not specify more details).
Simm et al. (2011) [59]	No cases were reported.	NDA	Dental evaluation.	N/A
Tessaris et al. (2016) [60]	No cases were reported.	NDA	Clinical and imaging evaluation (orthopantomography and CTCB).	NDA
Uday et al. (2018) [61]	1 case. MRONJ Stage 2 after extraction of a fractured molar.	NDA	Clinical evaluation.	Amoxicillin and metronidazole as a treatment for MRONJ.
Vuorimies et al. (2011) [62]	No cases were reported.	NDA	Not specific.	N/A
Wagner et al. (2011) [63]	No cases were reported.	NDA	Not specific.	N/A

ENT: ear, nose, throat; NDA: no data available.

**Table 3 jcm-12-01416-t003:** Distribution of data from patients undergoing invasive dental procedures and MRONJ report.

Authors	Number of pts.	Avg. Age (Range)	Sex	Medication	Invasive Procedures	Time of Invasive Procedure (during tx./After tx./Unknown)	Duration of ART tx. Prior to the inVasive Procedure (Avg. in yr.)	Cumulative Dose Prior to the Invasive Procedure	Use of Antibiotic Therapy	Post-Operatory Follow-up Time (Range) in yr.	MRONJ Report
Bredell et al. [36]	1	18.00	1(F)	DS	Maxillary exploratory surgery.	1/0/0	0.66	NDA	NDA	NDA	
Brown et al. [37]	11	6.61 (1.65 to 12.47)	NDA	PD and ZD	20 primary teeth extractions, 4 permanent teeth surgical extractions included, 2 simple permanent teeth extractions, 1 canine surgical exposure included, 1 odontoma excision.	28/0/0	PD (*n* = 11): 3.50 ZD (*n* = 6): 1.20	PD: 1250 mg on avg. ZD: 11.1 mg on avg.	NDA	NDA	1 case (stage 2).
Chahine et al. [39]	113	14.00 (2.00 to 30.00). (Data from only 66 patients).	41 (M) 25 (F) (Data from only 66 patients).	PD	178 primary tooth extractions, 72 permanent teeth extractions (32 simple and 40 surgical).	163/87/0	4.60 (Data from only 66 patients)	40 mg/kg on avg. (Data from only 66 patients).	Applied in 12 patients. (Data from only 66 patients).	0.23 (0.00 to 3.75). Telephone follow-up: 2.48 (0.12 to 10.38)	None
Ierardo et al. [43]	20	NDA (8.00 to 14.00)	12 (M) 8 (F)	ND	35 scaling (tart removal), 20 germenectomies, 15 primary tooth extractions, 5 canine extractions included, 8 frenilectomies, 3 ankylosed molar extractions.	86 / 0 / 0	NDA	NDA	Used with 2 patients. Amoxicillin 50 mg/kg 30–60 min before surgery. Clindamycin 20 mg/kg was used in allergy sufferers 30–60 min before surgery.	NDA (2 to 5)	None
Maines et al. [48]	2	NDA	NDA	ND	2 pulpectomies.	2/0/0	NDA	NDA	NDA	NDA	None
Malmgren et al. [49]	22	12.20 (3.40 to 31.90)	10 (M) 12 (F)	PD	30 tooth extractions, 6 surgical extractions, 1 orthognathic surgery, 1 implant surgery, 1 endodontic treatment.	33/6/0	3.60	NDA	NDA	3.10 (0.90 to 8.00)	None
Milano et al. [50]	1	4.66	1 (M)	PD	5 primary tooth extractions.	5/0/0	NDA	NDA	Not applied.	2.75	None
Ngan et al. [54]	1	12.00	1 (F)	PD	6 primary tooth extractions.	6 / 0 / 0	NDA	NDA	Amoxicillin for 5 days.	0.25	None
Okawa et al. [55]	31	NDA	NDA	NDA	67 primary tooth extractions.	67 / 0 / 0	NDA	NDA	NDA	NDA	None
Schwartz et al. [58]	13	NDA (2.00 to 16.00)	NDA	NDA	50 extractions of primary teeth and 10 of permanent teeth.	39 / 11 / 10	NDA	NDA	Used in 12 patients, not used with 2 and there is no data from 1 patient.	NDA	None
Uday et al. [61]	1	19.00	1 (M)	DS	1 permanent tooth extraction.	1/0/0	NDA	98 mg/kg	NDA	NDA	1 case (stage 2).

ART: antiresorptive; avg.: average; DS: denosumab; ND: neridronate; PD: pamidronate; pts.: patient(s); tx.: treatment; yr.: year(s); ZD: zoledronate.

**Table 4 jcm-12-01416-t004:** Assessment of methodological quality and risk of bias in the 29 articles included in this systematic review.

Cohort Study			
Authors	Verification List	Assessment	Interpretation
Brown et al. [37]	Checklist SIGN Methodology 3 (cohort studies)	Acceptable	Medium risk
Carpenter et al. [38]		High quality	Low risk
Chahine et al. [39]		Low quality	High risk
Feehan et al. [40]		High quality	Low risk
Idolazzi et al. [42]		High quality	Low risk
Johannesen et al. [44]		High quality	Low risk
Kumar et al. [45]		Acceptable	Medium risk
Li et al. [46]		Low quality	High risk
Lindahl et al. [47]		High quality	Low risk
Maines et al. [48]		High quality	Low risk
Malmgren et al. [49]		High quality	Low risk
Nasomyont et al. [53]		High quality	Low risk
Okawa et al. [55]		High quality	Low risk
Putman et al. [57]		High quality	Low risk
Voumiries et al. [62]		High quality	Low risk
Wagner et al. [63]		High quality	Low risk
Reports and case series			
August et al. [35]	Murad et al. Methodological quality and synthesis of case series and case reports	2/5	High risk
Bredell et al. [36]		5/5	Low risk
Ierardo et al. [43] Milano et al. [50]		1/5	High risk
		4/5	Medium risk
Moeini et al. [51]		0/5	High risk
Naidu et al. [52]		5/5	Low risk
Ngan et al. [54]		4/5	Medium risk
Schwartz et al. [58]		4/5	Medium risk
Simm et al. [59]		5/5	Low risk
Tessaris et al. [60]		5/5	Low risk
Uday et al. [61]		5/5	Low risk
Non-randomized clinical trials			
Goldsby et al. [41]	Sterne et al. ROBINS-I tool for assessing risk of bias in non-randomized studies of interventions	Low risk	Low risk
Randomized clinical trials			
Piperno-Neumann et al. [56]	Checklist SIGN Methodology 2 (controlled trials)	High quality	Low risk

## Data Availability

Data available by request: henryagg@gmail.com.

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
