# Peer review of "Medication-Related Osteonecrosis of the Jaws (MRONJ) in Children and Young Patients—A Systematic Review"

_jcm, 2023, doi:10.3390/jcm12041416_

Round 1

Reviewer 1 Report (Previous Reviewer 3)

I have previously suggested further changes to the authors to improve the text.

If the changes have been made following the suggestions of the other reviewers, the manuscript can be evaluated for publication.

In addition, I suggest including a specific section on the role of the surgical management in improving MRONJ at early stages with recent and adequate references.

Author Response

Rev.:  I have previously suggested further changes to the authors to improve the text. If the changes have been made following the suggestions of the other reviewers, the manuscript can be evaluated for publication.

Reply: All the review process was in agree with the reviewers following their suggestions and comments.

Rev.: In addition, I suggest including a specific section on the role of the surgical management in improving MRONJ at early stages with recent and adequate references.

Reply:

Dear reviewer thank you for your recommendations.

Even though the main  objective of the article is  epidemiological because of the lack of evidence concerning the MRONJ in children , and even less in  early stages surgery  in peadiatric patients, we will include a section on the topic the role of the surgical management in improving MRONJ in the discussion chapter

Reviewer 2 Report (New Reviewer)

The paper "Medicated Related Osteonecrosis of the Jaws (MRONJ) in Children 2 and Young Patients. A Systematic Review" is well written and provides important information. The review follows the Preferred Reporting Items for Systematic reviews and Meta-Analyses PRISMA statement guidelines. Generally, the manuscript is well-designed and well-described. However, there are some areas for improvement. Here below are my comments.

Results

Line 164. In studies by Moeini et al., Okawa et al., Schwartz et al. (ref. 51,55,58) it was not specified the involved drug. “Discussion” section (Line 401,402) “This fact makes it essential that studies using bisphosphonates include information about the drug and the cumulative doses, in order to contribute to establishing risk groups in these populations.”

Why these studies were not excluded?

Tables.

Please check that all abbreviations are present in legend. 

Conclusion

It should include future recommendations.

Author Response

Rev.:  The paper "Medicated Related Osteonecrosis of the Jaws (MRONJ) in Children 2 and Young Patients. A Systematic Review" is well written and provides important information. The review follows the Preferred Reporting Items for Systematic reviews and Meta-Analyses PRISMA statement guidelines. Generally, the manuscript is well-designed and well-described. However, there are some areas for improvement. Here below are my comments.

Reply: Thank you

Rev.: Results - Line 164. In studies by Moeini et al., Okawa et al., Schwartz et al. (ref. 51,55,58) it was not specified the involved drug. “Discussion” section (Line 401,402) “This fact makes it essential that studies using bisphosphonates include information about the drug and the cumulative doses, in order to contribute to establishing risk groups in these populations.” Why these studies were not excluded?

Reply:

Thank you for the interesting question :

The scant literature available where child and adolescent populations are under antiresorptive treatment with medication makes it necessary to include the aforementioned studies, with the aim of increasing the population of this systematic review. For this reason, the exclusion criteria do not contemplate the discarding of articles with certain deficiencies in pharmacological data.

Our inclusion criteria  was as follows  ¨ The following inclusion criteria were considered: articles in English and Spanish, with child-youth population, undergoing treatment with antiresorptive and/or antiangiogenic using intravenous route (IV) or oral route (po), with reports of cases of MRONJ. Case reports and case series, prospective and retrospective associated studies, case-control studies, randomized and non-randomized clinical trials were included. The exclusion criteria were literature review, treatment in subjects with more than 18 years old and the inability to access the full text. ¨

For the information presented above those mentioned articles were not excluded , even though the discussion indicates the importance of  the information regarding the drugs, that is why in the recommendations of the article better  standardization of the information is indicated.

Rev.: Tables - Please check that all abbreviations are present in legend. 

Reply: thank you for the recommendations dear reviewer  all the abreviations has been check  and marked on the text  legends

Rev.: Conclusion - It should include future recommendations.

Reply:  thank you for the recommendation ,  this topic has been added to the text and marked .

Round 2

Reviewer 1 Report (Previous Reviewer 3)

Authors improved the manuscript quality.

I suggest to refer to this important manuscript, cited in the last MRONJ update by AAOMS. [https://doi.org/10.1016/j.joms.2020.05.037]

Author Response

Dear reviewer, thank you for your comment.

The reference cited by you, indeed is a very complete reference. The reference [https://doi.org/10.1016/j.joms.2020.05.037]: 
Giudice A, Barone S, Diodati F, et al: Can surgical management improve resolution of medication-related osteonecrosis of the jaw at early stages? A prospective cohort study. J Oral Maxillofac Surg. 2020; 78:1986-1999.
Is included in the manuscript in the number 106 and was included in the discussion chapter in the last review process. Giudice et al. have another articles included into the paper as well.

This manuscript is a resubmission of an earlier submission. The following is a list of the peer review reports and author responses from that submission.

Round 1

Reviewer 1 Report

In general:

No table could be viewed in the PDF. Have they been properly uploaded to the site?

Prefer the number of articles to the percentage. For example, write 16 articles (55.17%) instead of 55.17% of the articles (n=16). Harmonize this throughout the text 

Please consider the AAOMS update released in 2022: DOI: 10.1016/j.joms.2022.02.008

Section materials and methods :

"In the second round of search and evaluation, a manual selection of references in all articles included after the first round." => please rephrase

Section results : 

In flowchart, add 2417 excluded articles after removing duplicate

Section Osteonecrosis of the jaws:

Correct redundancies in the number of patients. Don't write "in seven 7 patients" but in seven patients or in 7 patients. Harmonize this throughout the text 

Section Invasive procedure : 

What is the different between permanent tooth surgical extractions and simple permanent tooth surgical extractions?

"In one article (11.11%), 2.48 years of telephone postoperative follow-up was reported [39]": please rephrase.

You can add this article in your review: DOI: https://doi.org/10.1016/j.ijom.2011.07.606

Section discussion:

"In patients with osteoporosis exposed to antiresorptive, the prevalence is calculated at 0.1%, observing an increase of the same to 0,21% in the population with more than 4 years of treatment" : please rephrase

Author Response

Dear reviewer

Thank you for your review and your help. All of your comment was added as follow

Rev.: Prefer the number of articles to the percentage. For example, write 16 articles (55.17%) instead of 55.17% of the articles (n=16). Harmonize this throughout the text 

Reply: this was modified.

Rev.: Please consider the AAOMS update released in 2022: DOI: 10.1016/j.joms.2022.02.008

Reply: Thank you, this article was included in the discussion chapter.

Rev.: "In the second round of search and evaluation, a manual selection of references in all articles included after the first round." => please rephrase

Reply: the sentence was modified: “In the second round of search, a manual analysis of the references in each article was realized”

Rev.: In flowchart, add 2417 excluded articles after removing duplicate

Reply: thank you; the flowchart was modified.

Section Osteonecrosis of the jaws:

Rev.: Correct redundancies in the number of patients. Don't write "in seven 7 patients" but in seven patients or in 7 patients. Harmonize this throughout the text 

Reply: thank you; this was changed

Section Invasive procedure: 

Rev.: What is the different between permanent tooth surgical extractions and simple permanent tooth surgical extractions?

Reply: you are right. The number of cases was included in the same concept “permanent tooth surgical extraction”

Rev.: "In one article (11.11%), 2.48 years of telephone postoperative follow-up was reported [39]": please rephrase.

Reply: thank you; the sentence was modified to: “In one article (11.11%) with 2.48 years follow-up, the postoperative control was performed calling all the subjects or parents”

Rev.:  You can add this article in your review: DOI: https://doi.org/10.1016/j.ijom.2011.07.606

Reply: thank you for your comment; the article related to DOI is “Orthognathic surgery in a patient with osteogenesis imperfecta: a case study”, by Fariña R and Pérez Araya J. This is an abstract publishing in the IJOMS showing an experience with a patient; we was treating to find the full text paper related to this abstract but was no possible to find; if you can help us in this search, we can add the article. Thank you!

Section discussion:

Rev.:  "In patients with osteoporosis exposed to antiresorptive, the prevalence is calculated at 0.1%, observing an increase of the same to 0,21% in the population with more than 4 years of treatment" : please rephrase

Reply: thank you; the sentence was modified to: “In patients with osteoporosis, the prevalence is 0.1% with an increase to 0,21% when the use of the drug is for more than 4 years”

Reviewer 2 Report

Many thanks for this elaborate systematic review. The work to review all the studies on the subject was certainly very large and time-consuming.

For easier review, page numbers and line numbers would be helpful. The tables were not included in the manuscript/not accessible and therefore could not be considered.

Literature on the prevalence of MRONJ in children and adolescents is very limited and available data quality is poor as presented in this systematic review. Data included consists mainly of case series and cohort studies (>90%). Even basic information such as sex are missing in 25% and age in 10% of the patients. Furthermore, there is a huge heterogeneity of age (range 0-32 years) with therefore completely different bone metabolism and likely MRONJ risk.

Why not focus only on OI patients (60%)? Why not limit your collective (e.g. age < 18 years) to make a possible valid statement?

Unfortunately, the quality of the studies is so poor that, of course, no valid conclusions can be drawn from this elaborate systematic review. In the end, the assumption remains that the prevalence of MRONJ in children and young adults seems to be low.

The conclusions should therefore be expressed more conservatively, e.g.

P.12 …concomitant medication does not APPEAR to be a risk factor …

P.12 … invasive treatments in this population do not APPEAR to be a risk factor …

...

Author Response

Dear reviewer

Thank you for your comment and suggestions. All of your comment was added as follow.

Rev.:  Many thanks for this elaborate systematic review. The work to review all the studies on the subject was certainly very large and time-consuming.

Reply: thank you for your comments

Rev.:  Literature on the prevalence of MRONJ in children and adolescents is very limited and available data quality is poor as presented in this systematic review. Data included consists mainly of case series and cohort studies (>90%). Even basic information such as sex are missing in 25% and age in 10% of the patients. Furthermore, there is a huge heterogeneity of age (range 0-32 years) with therefore completely different bone metabolism and likely MRONJ risk.

Reply: you are right, and we are in agree with your statement; this is one of the aims of this research, showing the limited data related to MRONJ and the urgent requirements for good quality research

Rev.:  Why not focus only on OI patients (60%)? Why not limit your collective (e.g. age < 18 years) to make a possible valid statement? Unfortunately, the quality of the studies is so poor that, of course, no valid conclusions can be drawn from this elaborate systematic review. In the end, the assumption remains that the prevalence of MRONJ in children and young adults seems to be low.

Reply: thank you for your comment and its right; those position was a choice at the beginning of this proposal research; however, as 1) data are poor, 2) the patients included in different articles show a mix in ages, 3) diagnosis included are variables and 4) protocols for the use of drugs show differences in almost each included articles, the final decision was to perform the present article including data from different papers and to treat to create a classification as well as possible with the systematic review strategy.

Rev.:  The conclusions should therefore be expressed more conservatively, e.g.

P.12 …concomitant medication does not APPEAR to be a risk factor …

P.12 … invasive treatments in this population do not APPEAR to be a risk factor …

Reply: the sentences were modified as follow:

“that reason the use of concomitant medication does not appear to be a risk factor in the development of MRONJ.”

“These data allow us to clarify that invasive treatments in this population do not appear to be a risk factor for the MRONJ.”

Reviewer 3 Report

Manuscript ID: jcm-1890817

Type: Systematic review

Title: Medicated Related Osteonecrosis of the Jaws (MRONJ) in Children and Young Patients. A Systematic Review

Authors must draft their manuscript following the template provided by the journal.

Abstract: "Medication related osteonecrosis of the jaw (MRONJ) is defined by the American Association of Oral and Maxillofacial Surgeons (AAOMS) as the presence of an exposed bone area or the possibility of probing bone tissue through an intraoral or extraoral fistula in the maxillofacial region, which persists for more than eight weeks, in patients with current or previous treatment with antiresorptive or antiangiogenic agents, with no history of radiation or metastatic disease in the maxillofacial region."
Reduce this part.

Review: I suggest not including manuscripts in Spanish.

The systematic review could be influenced by this.

Reference: some of the reference are very dated (n. 2 - 4 - 5 - 7 - 10 - 11 - 17 - 18 - 19). Given the fairly common topic in the literature, I strongly suggest replacing them with some more recent ones [DOI10.1016/j.joms.2020.05.037 - doi: 10.1016/j.oooo.2018.09.008 - doi: 10.1016/j.bone.2019.04.010 - DOI10.4103/ijdr.IJDR_689_19 - doi: 10.1620/tjem.247.75 - doi: 10.4317/medoral.23191].

This helps to improve the scientific soundness of the manuscript.

After the changes have been made, one can start carefully revising the manuscript.

I advise authors to review the manuscript in full, making the necessary changes to proceed with the systematic review.

Author Response

Thank you for your review and your help. All of your comment was added and your comment about references was full included in this review. 

Rev.:  Abstract: "Medication related osteonecrosis of the jaw (MRONJ) is defined by the American Association of Oral and Maxillofacial Surgeons (AAOMS) as the presence of an exposed bone area or the possibility of probing bone tissue through an intraoral or extraoral fistula in the maxillofacial region, which persists for more than eight weeks, in patients with current or previous treatment with antiresorptive or antiangiogenic agents, with no history of radiation or metastatic disease in the maxillofacial region."

Reduce this part.

Reply: the sentences were modified as follow: “Medication related osteonecrosis of the jaw (MRONJ) is defined by the American Association of Oral and Maxillofacial Surgeons (AAOMS) as the presence of an exposed bone area in the maxillofacial region, present for more than eight weeks in patients with use of antiresorptive or antiangiogenic agents, with no history of radiation or metastatic disease”

Rev.: suggest not including manuscripts in Spanish. The systematic review could be influenced by this.

Reply: the search was performed using English and Spanish, however, no article was founded in Spanish; we can delete the Spanish in the strategy. Was deleted articles in Spanish from the references and was change to scientific literature in English as well.

Rev.:  some of the references are very dated (n. 2 - 4 - 5 - 7 - 10 - 11 - 17 - 18 - 19). Given the fairly common topic in the literature, I strongly suggest replacing them with some more recent ones

[DOI10.1016/j.joms.2020.05.037 – 

doi: 10.1016/j.oooo.2018.09.008 – 

doi: 10.1016/j.bone.2019.04.010 – 

DOI10.4103/ijdr.IJDR_689_19 – 

doi: 10.1620/tjem.247.75 –

doi: 10.4317/medoral.23191].

This helps to improve the scientific soundness of the manuscript.

Reply: thank you. The references were reviewed and changed. Were included theses new references and deleted the others as you suggested.

Round 2

Reviewer 2 Report

Many thanks for revising your work. All my points were answered.

I consider the finding of this systematic review to be moderate, because the data situation hardly allows reliable conclusions. It is not a surprising finding that better studies are needed to make a statement when the data consist more or less of case series and cohort studies.

Author Response

Rev.:  Many thanks for revising your work. All my points were answered.I consider the finding of this systematic review to be moderate, because the data situation hardly allows reliable conclusions. It is not a surprising finding that better studies are needed to make a statement when the data consist more or less of case series and cohort studies.

Reply: thank you for your comments; we performed a systematic review using the best evidence at the moment and this facts was included in the main text.

Reviewer 3 Report

Authors modified the manuscript however, the manuscript must to be  improved in many parts. 
Authors must draft their manuscript following the template provided by the journal. They must download the template from the Journal website.
References must to be inserted following Mendeley method.
"This increase in potency, and in turn in toxicity, is due to the presence of nitrogen within molecular chain of bisphosphonate, and due to the substitution of carbon for oxygen within its chemical structure, bisphosphonates are completely resistant to hydrolysis, which explains its accumulation in bone tissue and its long half-life, calculated at more than 11 years"
I suggest to add these relevant recent references on the mechanism of bisphosphonates [DOI: 10.3390/ph13120423 -http://doi.org/10.4065/83.9.1032 -  DOI: 10.1177/0963689720948497]

After the revision, manuscript must be re-evaluated.

Author Response

Rev.:  "This increase in potency, and in turn in toxicity, is due to the presence of nitrogen within molecular chain of bisphosphonate, and due to the substitution of carbon for oxygen within its chemical structure, bisphosphonates are completely resistant to hydrolysis, which explains its accumulation in bone tissue and its long half-life, calculated at more than 11 years"

Reply: the paragraph was modified

Rev.: I suggest to add these relevant recent references  [DOI: 10.3390/ph13120423 -http://doi.org/10.4065/83.9.1032 -  DOI: 10.1177/0963689720948497]

Reply: the articles were included in the main text.